# Could Endogenous Glucocorticoids Influence SARS-CoV-2 Infectivity?

**DOI:** 10.3390/cells11192955

**Published:** 2022-09-21

**Authors:** Eugenio Hardy, Carlos Fernandez-Patron

**Affiliations:** 1Center of Molecular Immunology, P.O. Box 16040, Havana 11600, Cuba; 2Department of Biochemistry, Faculty of Medicine and Dentistry, College of Health Sciences, University of Alberta, Edmonton, AB T6G 2H7, Canada

**Keywords:** COVID-19, SARS-CoV-2, dexamethasone, cortisol, glucocorticoids, spike protein, ACE2 (angiotensin converting enzyme 2), innate immunity

## Abstract

Endogenous glucocorticoids and their synthetic analogues, such as dexamethasone, stimulate receptor-mediated signal transduction mechanisms on target cells. Some of these mechanisms result in beneficial outcomes whereas others are deleterious in the settings of pathogen infections and immunological disorders. Here, we review recent studies by several groups, including our group, showing that glucocorticoids can directly interact with protein components on SARS-CoV-2, the causative agent of COVID-19. We postulate an antiviral defence mechanism by which endogenous glucocorticoids (e.g., cortisol produced in response to SARS-CoV-2 infection) can bind to multiple sites on SARS-CoV-2 surface protein, Spike, inducing conformational alterations in Spike subunit 1 (S1) that inhibit SARS-CoV-2 interaction with the host SARS-CoV-2 receptor, ACE2. We suggest that glucocorticoids-mediated inhibition of S1 interaction with ACE2 may, consequently, affect SARS-CoV-2 infectivity. Further, glucocorticoids interactions with Spike could protect against a broad spectrum of coronaviruses and their variants that utilize Spike for infection of the host. These notions may be useful for the design of new antivirals for coronavirus diseases.

## 1. Introduction

The classical view of the mechanism of action of endogenous glucocorticoids is that these molecules, once released from adrenal cortex, primary lymphoid organs, the intestine, skin, or brain, circulate in the blood stream thus impacting systemic cellular immunity, inflammatory signaling, and the metabolism of lipids and carbohydrates via glucocorticoid receptor-mediated signal transduction pathways on target cells [1,2].

The impact of glucocorticoids on cellular immunity is complex and target cell type specific [3,4,5]. In epithelial cells and antigen-presenting cells such as macrophages and dendritic cells, glucocorticoids can inhibit the production of inflammatory cytokines and chemokines. In effector cells such as cluster of differentiation 8 positive (CD8^+^) T lymphocytes, in type-1 helper T cells, and natural killer cells, glucocorticoids signaling can downregulate IFN-γ expression, which may impair cell-mediated immune responses. These effects protect target cells from exaggerated inflammation at the cost of an increased cellular susceptibility to pathogen infections under conditions that stimulate glucocorticoids production, such as physiological or psychological stress [3,4,5].

The strong anti-inflammatory and immune modulatory actions of endogenous glucocorticoids has been the basis for the design of structurally analogous synthetic drugs to treat patients’ allergies, asthma, autoimmunity, cancer as well as infections caused by pathogens [5,6]. Several synthetic glucocorticoids such as dexamethasone, prednisolone, and prednisone have been widely tested for the treatment of viral influenza, and coronavirus infections such coronavirus disease of 2019 (COVID-19) and severe acute respiratory syndrome and middle east respiratory syndrome with varied efficacies [6,7,8,9,10]. Recently, dexamethasone (6 mg/d for 10 days) has been found to be beneficial for preventing one in three deaths among patients receiving invasive mechanical ventilation, and one death in five for patients on oxygen [11]. Dexamethasone does not produce significant benefit in some patients such as those who do not receive respiratory support, according to the Randomized Evaluation of COVID-19 Therapy trial conducted in United Kingdom [11]. The mechanisms by which low doses of dexamethasone reduce mortality in some patients with severe COVID-19 disease are not fully understood, making it difficult to improve dexamethasone efficacy and application to a broader spectrum of SARS-CoV-2 infected patients. 

It has been suggested that, in some COVID-19 patients, dexamethasone acts via glucocorticoid receptors and the resulting intracellular signaling pathways, producing genomic responses that attenuate inflammation [12], while dexamethasone nongenomic effects warrant further consideration. 

Recent research including ours (Hassan et al. [13]) suggests that glucocorticoids, such as dexamethasone and cortisol, can directly interact with molecular targets unrelated to the glucocorticoids receptors such as the SARS-CoV-2 Spike subunit 1 (S1) protein in ways that may reduce viral infectivity. We hypothesize that endogenous glucocorticoids may play a new innate immunity role in protecting the human host against infection with SARS-CoV-2, and possibly other coronaviruses that use Spike to enter into host cells.

## 2. Various Lines of Research Suggest That Endogenous Glucocorticoids and Synthetic Analogs Thereof May Impact SARS-CoV-2 Infectivity by Interacting with the Host and Viral Components

### 2.1. Dexamethasone May Interact with the Host to Reduce ACE2 Expression

A study by Shahbaz et al. [14] has shed some light on why dexamethasone treatment may benefit individuals with low blood-oxygen levels in which SARS-CoV-2 has infected and destroyed immature red blood cells. The authors found that red blood cells precursors/progenitors [CD71^+^ Erythroid Cells (CECs)] co-expressing angiotensin-converting enzyme 2 (ACE2), a major host receptor for coronavirus, and the SARS-CoV-2 co-receptor (TMPRSS2) are enriched in the blood of COVID-19 patients and that they support viral replication. Further, CECs pre-treated with dexamethasone for 24 h before infection with SARS-CoV-2 exhibit diminished ACE2 expression and reduced SARS-CoV-2 infectivity. These interactions of dexamethasone with host cells are presumably mediated by the glucocorticoids receptors. The authors conclude that dexamethasone may help to decrease mortality and disease duration in hypoxic COVID-19 patients by reducing the susceptibility of their immature red blood cells to be infected by SARS-CoV-2 [14]. 

Using computational docking and molecular dynamics simulations, Fadaka et al. [15] found that dexamethasone can bind to host secreted proteins such as the highly pro-inflammatory cytokine, interleukin-6. Accordingly, dexamethasone binds to interleukine-6 through establishing hydrogen bonds with three amino acids (Q175, D34 and L33), as well as via ionic, salt bridges and hydrophobic contacts that are needed for the correct folding, stability and activity of interleukin-6 [15]. The consequences of dexamethasone binding to interleukin-6 function remain to be explored.

### 2.2. Dexamethasone May Directly Interact with SARS-CoV-2 Components

Fadaka et al. [15] computational docking and molecular dynamics simulations data has further indicated that dexamethasone may influence COVID-19 disease development through binding to its glucocorticoid host receptor as well as via interactions with the main SARS-CoV-2 protease (3-chymotrypsin-like protease, M^pro^) which participates in enabling viral replication through the conversion of polyproteins such as pp1a and pp1ab to functional proteins. 

In other in silico studies, Ghosh et al. [16] similarly concluded that dexamethasone can interact with many amino acids of M^pro^ including catalytic (H41 and C145) residues, with a binding affinity of −7.9 kcal/mol. These authors suggested that dexamethasone could be more effective in inhibiting SARS CoV-2 M^pro^ activity than two anti-HIV drugs that interact with M^pro^ (darunavir with a binding affinity of −7.4 kcal/mol and lopinavir with a binding affinity of −7.3 kcal/mol) [16]. That dexamethasone can target the binding cavity of M^pro^ has been confirmed by other researchers using computational techniques [17,18,19]. Dexamethasone interactions with M^pro^ may involve ionic contact, hydrogen bond, van der Waals interactions, and hydrophobic interactions [16,20].

In addition to M^pro^, Shoemark et al. [21] examined, by means of docking and molecular dynamics simulations, the properties of the SARS-CoV-2 Spike trimer in open and locked conformations and the binding of dexamethasone to a fatty acid binding site on SARS-CoV-2 Spike protein. These authors found that, among a varied population of Spike structural states [22], dexamethasone can be accommodated in the fatty acid binding site and stabilizes the locked Spike trimer structure at least during the timescales (200 ns) used for their molecular dynamics simulations [21]. The fatty acid pocket in SARS-CoV-2 Spike trimer can bind linoleic acid (a polyunsaturated omega-6 fatty acid) which generates and stabilizes a locked (compacted) conformation of the receptor-binding domain (RBD) trimer and interferes in vitro with Spike binding to ACE2 [22]. While Spike conformations (such as open Spike trimer) are accessible for ACE2 receptor interaction, the dexamethasone-generated locked Spike with an occluded RBD surface should exhibit reduced affinity for ACE2 [21], which could thus affect SARS-CoV-2 infectivity. Studies by our group [13] may lend experimental support to the analyses by Shoemark et al. [21].

### 2.3. Dxamethasone and Cortisol Bind to Multiple Sites on SARS-CoV-2 S1 Protein to Cooperatively Inhibit S1-ACE2 Interaction

We [13] and other investigators [21] have recently tested a variety of small molecule ligands including glucocorticoids (such as dexamethasone and cortisol) in their ability to interact with SARS-CoV-2 Spike protein, generating altered conformations of Spike with reduced binding affinity for ACE2. Both dexamethasone and cortisol may stably interact with multiple high affinity pockets in SARS-CoV-2 S1 (Table 1, row 1), a result we confirmed using complementary, redundant and orthogonal biochemical approaches including limited proteolysis of S1 in the presence of cortisol followed by mass spectrometric identification of cortisol binding peptides that we predicted by using molecular dynamics simulations and other in silico studies (Table 1, rows 2–4) [13]. The data suggest that binding of dexamethasone or cortisol to SARS-CoV-2 S1 causes S1 to partially unfold, thus disrupting the binding of SARS-CoV-2 S1 to ACE2 (Table 1, row 3). 

Based on our data, dexamethasone and cortisol are unlikely to disturb the catalytic site in ACE2 as they do not compromise ACE2 enzymatic function [13]. However, it is possible that these glucocorticoids can bind to other sites on ACE-2. Zhang et al. [23] studied the binding of dexamethasone to SARS-CoV-2 RBD and ACE2 by means of molecular docking, surface plasmon resonance and bioaffinity chromatography equipped with an ACE2^h^-cell membrane chromatography column. These authors found that dexamethasone interacts (presumably forming hydrogen bonds) with amino acids Q498 and Q496 from the RBD as well as K353 and K354 from the ACE2 receptor. The amino acids Q498 and K353 are active sites for RBD binding to ACE2 during SARS-CoV-2 infection [23]. According to these authors, dexamethasone interacts with ACE2 with a binding constant (K_D_) of (9.03 ± 0.78)10^−6^ M. Moreover, dexamethasone (10 μM) may block (~41 %) the entry (into ACE^h^ cells) of SARS-CoV-2 Spike pseudotyped virus (a model of SARS-CoV-2 viropexis) [24]. These authors proposed that dexamethasone blocks ACE2 binding to SARS-CoV-2 Spike protein.

Interestingly, we found cortisol to concentration-dependently disrupt the interaction between the SARS-CoV-2 S1 Beta variant (E484K, K417N, N501Y) and ACE2 [13]; 100 nM cortisol results in ~55% inhibition. By contrast, specific mutations in the Delta and Omicron variants, which are in or in the vicinity of cortisol- and dexamethasone-binding pockets, may affect the binding and affinity of the glucocorticoids to S1 (Table 1, row 2). Accordingly, Spike mutations that weaken the inhibition by cortisol or dexamethasone of SARS-CoV-2 S1-ACE2 interactions could increase viral infectivity.

These findings suggest that endogenous glucocorticoids may play new, potentially important roles in innate immunity through their ability to directly interact with SARS-CoV-2, in addition to their classical mode of action mediated by glucocorticoid receptors which are expressed in almost all cells in the body [1].

Interestingly, mixing dexamethasone and cortisol can be more effective at inhibiting the S1-ACE2 interaction than dexamethasone or cortisol alone, by effect of their cooperative binding to SARS-CoV-2 S1 (Table 1, row 4) [13]. A similar cooperative inhibition of the biochemical S1-ACE2 interaction is exhibited by cortisol and antibodies against the receptor binding domain of S1 (Table 1, row 4). Whether endogenous glucocorticoids and S1 antibodies can cooperatively reduce viral infectivity in SARS-CoV-2 infected individuals warrants investigation. 

## 3. Baseline Levels of Endogenous Glucocorticoids and SARS-CoV-2-Induced Glucocorticoid Overproduction May Influence SARS-CoV-2 Interaction with ACE2

### 3.1. Glucocorticoids Constitutively Secreted by Adrenal Glands May Be Available for a Direct Interaction with SARS-CoV-2 Spike Immediately after Infection of the Host

The innate immune system is an organism’s response to insults that can range from an infection by pathogens (viruses, bacteria and parasites) to organism’s damage due to trauma, wounds or disease conditions [25,26]. In humans, the innate immune system can be understood as the physical barriers (skin, respiratory tract, gastrointestinal tract), secretions (saliva, gastric acid, mucous) and the biological responses (inflammation, complement) to insults originating in (self) or outside (non-self) the organism [27,28]. The innate immune system may be viewed as an organ of perception. Molecular perception is in, part provided, by a spectrum of pattern recognition receptors which include Toll-like receptors, Nod-like receptors, Rig-like receptors, absent in melanoma 2-like receptors, and C-type lectin receptors [25,26,27,28]. Potent innate immune responses generally occur when these pattern recognition receptors become activated by ligands termed pathogen-associated molecular patterns and damage-associated molecular patterns. On one hand, the ensuing innate immunity responses provide the organism with a defence mechanism against the initial insults. On the other hand, uncontrolled receptor-mediated immune responses can cause inflammatory damage that is detrimental (immunopathological) to the organism [25,26,27,28]. 

Innate immune responses that proceed without the engagement of pattern recognition receptors can provide immediate control of infection and danger with limited inflammation and immunopathology, as recently reviewed [27].

Glucocorticoids constitutively secreted under physiological (unstressed) conditions by the adrenal glands (Figure 1, left panel) regulate systemic metabolism, cardiovascular function, reproduction, development, growth, and immunity [1]. The release of glucocorticoids from the adrenal glands in the circulation exhibits circadian and ultradian rhythms [1]. In humans, cortisol is released in the bloodstream at different times during the sleep-wake cycle, typically peaking in the early morning and decreasing during the day. Our data [13] suggest that blood cortisol levels (which normally oscillate from 80 nM to 700 nM) could inhibit (at least partially) SARS-CoV-2 S1 interaction with ACE2 which could, consequently, impact SARS-CoV-2 infectivity.

### 3.2. Glucocorticoids Inhibition of SARS-CoV-2 S1 Interaction with ACE2 May Be a Constitutive/Inducible Innate Immunity Mechanism

Whether baseline levels of endogenous glucocorticoids, owing to their potential for binding to SARS-CoV-2 S1, serve an antiviral damage-limiting immune function in reducing SARS-CoV-2 infectivity is unknown and merits investigation. 

We propose that glucocorticoids binding to multiple sites in SARS-CoV-2 S1 may have several consequences:(i)S1 conformational changes may be induced which increase the rigidity of Spike protein reducing its affinity for ACE2.(ii)S1 may adopt ACE2-inaccessible conformational states (in contrast to ACE2-accessible conformational states, such structures have been proposed to not involve canyons between RBDs [22]), which would make SARS-CoV-2 less infectious.(iii)The affinity for ACE2 may be reduced for S1 structures having pockets occupied by glucocorticoids.(iv)SARS-CoV-2 infectivity may be significantly reduced (Figure 1).

Taken together, we propose a model (Figure 1) that could be classified as a constitutive/inducible innate immune response mediated by endogenous glucocorticoids according to previously defined criteria [27,32]. The constitutive component of this innate immune response would depend on the availability of a pool of the glucocorticoids normally produced and secreted under physiological conditions in a circadian fashion. This pool of glucocorticoids directly engages the pathogen weakening the pathogen’s potential to interact with the host. The inducible component would depend on glucocorticoids produced and secreted by host cells in response to the pathogen (e.g., SARS-CoV-2). The resultant protection of the host would take place without engagement of the classical host’s glucocorticoid receptors, antigen-specific receptors or pattern recognition receptors. Conceivably, the postulated constitutive/inducible innate immune response mediated by glucocorticoids would act against danger (SARS-CoV-2) in a largely non-inflammatory manner. 

A cytokine signaling cascade may, however, underlie the induction of glucocorticoids by SARS-CoV-2. It is known that glucocorticoids levels rise during physiological and psychological stress [3,4]. In response to SARS-CoV-2 infection, the production of early innate proinflammatory cytokines such as interleukin-1, -6, -8, and -12 and tumor necrosis factor-α, and interferons-α/β has been shown to be stimulated leading late (T-lymphocyte) cytokines such as INF-γ and IL-2 to be released [33]. Several of the proinflammatory cytokines; especially IL-1, which is very potent; interact with cytokine receptors located at all hypothalamic-pituitary-adrenal axis levels, resulting in dysregulated (e.g., enhanced) glucocorticoids (e.g., cortisol) production [34,35]. The hypothalamic paraventricular nucleus secretes corticotropin-releasing hormone and arginine vasopressin that stimulate the downstream secretion of adrenocorticotropic hormone from the pituitary gland into the blood circulation. Adrenocorticotropic hormone next stimulates the adrenal cortex, which results in the synthesis and secretion into the circulation of high levels of glucocorticoids (mainly cortisol in humans). The release of cytokines at each level of the hypothalamic-pituitary-adrenal axis and the direct effect of cytokines on the adrenal and pituitary glands, may maintain the glucocorticoid response [33]. Glucocorticoids overproduced by effect of SARS-CoV-2 infection could further contribute to inhibiting the interaction between SARS-CoV-2 S1 and the human SARS-CoV-2 receptor, ACE2.

## 4. Conclusions

Glucocorticoids have the capacity to interact with SARS-CoV-2 and their binding to high affinity pockets on SARS-CoV-2 S1 may lead to a cooperative inhibition of S1 binding to ACE2. Mutations in SARS-CoV-2 variants (such as Delta and Omicron) may interfere with glucocorticoids binding to S1. We suggest that glucocorticoids-mediated inhibition of S1 interaction with ACE2 may serve an innate immune function that protects against a broad spectrum of coronaviruses and their variants that utilize Spike for infection of the host. These notions may be useful for the design of new antivirals for coronavirus diseases.

## 5. Outlook 

Further research would be required to explore the application of the above-described constitutive/inducible glucocorticoids-mediated innate defence (antiviral) mechanism for prevention and treatment of current or future coronaviruses. We expect similar effects, as described here, of glucocorticoids on S1 from other coronaviruses (such as MERS-CoV and SARS-CoV) that use Spike protein for infection; this should be addressed through binding and structural studies. It could be that “hitting” coronaviruses including SARS-CoV-2 in various regions of Spike S1, with cocktails of glucocorticoids (such as dexamethasone, cortisol or the major mineralocorticoid aldosterone) or glucocorticoids and other compounds (such as regulatory approved triterpenoids, bile acids, or S1 neutralizing antibodies) results in cooperatively inducing conformational changes that preclude optimal S1 binding to ACE2 receptor. Unfavorable conformational changes may thereby significantly reduce coronavirus infectivity. If proven valid, this approach will contrast with the conventional pharmacological strategy (i.e., aiming for a highly specific ‘magic bullet’). A desirable feature of such antiviral cocktails is that they be applicable at low concentrations without toxicity, allowing extension of glucocorticoids-based treatments to a broad spectrum of coronavirus-infected patients.

Several questions are worthy of further investigation: (a)Could the proposed constitutive innate defence mechanism involving endogenous glucocorticoids contribute to asymptomatic COVID-19?(b)Could glucocorticoids have influenced the infectivity of some SARS-CoV-2 variants of concern?(c)Could blood factors such as plasma proteins acting as glucocorticoid transporters (e.g., corticosteroid-binding globulin, albumin, sex hormone-binding globulin) affect the binding of glucocorticoids to S1?(d)Do glucocorticoids act in concert with other ligands potentially capable of binding to S1, such as dietary polyunsaturated fatty acid, immunoreactive antibodies or even non-antibodies (proteins, peptides) in the blood, to shift the equilibrium towards stable, locked SARS-CoV-2 S1structures that prevent virus infectivity?(e)Do endogenous glucocorticoids serve a new innate immune function that protects against a broad spectrum of coronaviruses and their variants that utilize Spike for infection of the host?These notions may be useful for the design of new antivirals for coronavirus diseases.

## Figures and Tables

**Figure 1 cells-11-02955-f001:**
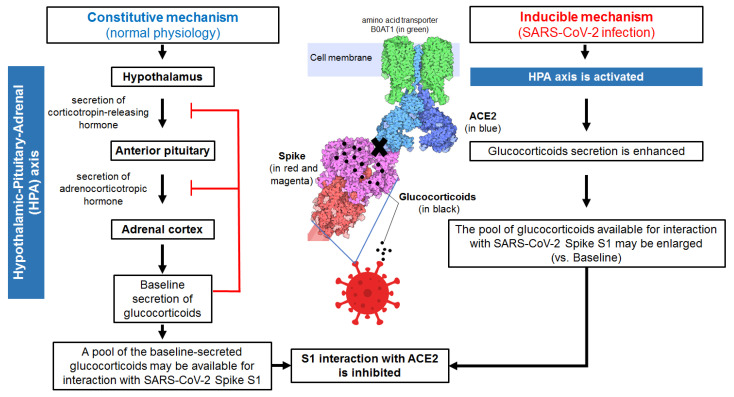
Schematic overview of the proposed innate immune mechanism (non-specific and non-genomic) by which glucocorticoids may reduce SARS-CoV-2 infectivity. Constitutive secretion of glucocorticoids downstream of the hypothalamic-pituitary-adrenal glands axis contributes to a baseline pool of glucocorticoids that may be available to interact with SARS-CoV-2 Spike 1 immediately after viral invasion. Direct binding of glucocorticoids to SARS-CoV-2 S1 results in altered (possibly locked) conformational states of the Spike protein. This mechanism could inhibit SARS-CoV-2 interaction with its main receptor in humans, ACE2. Under normal conditions, the constitutive secretion of glucocorticoids by the hypothalamic-pituitary-adrenal axis is regulated by end-product feedback inhibition, which limits the defensive potential of this constitutive innate immune response mechanism. However, SARS-CoV-2 infection itself is a stimulus for the hypothalamic-pituitary-adrenal axis leading to enhanced secretion of glucocorticoids from the adrenal cortex. As the secreted pool of glucocorticoids available for inhibition of S1 interaction with ACE2 increases, the antiviral defence potential of this viral induced innate immune mechanism may increase significantly. Note: Illustration of a complex of the S protein colored in red and magenta linked to ACE2 receptor colored in blue (https://cdn.rcsb.org/pdb101/motm/246/246-Coronavirus_Spike-6vsb_6m17_2.tif) was obtained from David Goodsell (https://doi.org/10.2210/rcsb_pdb/mom_2020_6), licensed under CC BY 4.0 (https://creativecommons.org/licenses/by/4.0/). The original illustration was created using PDB entries 6vxx [29], 6m17 [30], and 6vsb [31].

**Table 1 cells-11-02955-t001:** Summary of Results (from [13]) Showing that Glucocorticoids Bind to SARS-CoV-2 S1 and Block S1/ACE2 Interaction.

Row	Method(s)	Authors’ Main Results and Conclusions
1	Fpocket algorithm; Molecular docking using AutoDockVina and AutoDockTools; Molecular dynamics using GPU-accelerated AMBER, PyRED server, SHAKE algorithm, VMD, Ligplot, and PyMOL; Binding energy calculations, based on MM/GBSA	(i) In silico identification of 52 unique pockets, with high affinity for cortisol and dexamethasone, which are located and distributed on the RBD, NTD, RBD-RBD interface and NTD-RBD interface. (ii) The pockets interact with dexamethasone and cortisol with different affinity. (iii) The specificity of the pocket for dexamethasone or for cortisol depends on the unique glucocorticoid side chain-pocket interactions and the resultant binding affinity.
2	Limited proteolysis-coupled LC-MS (and methods in Row 1)	(i) Confirmation of several cortisol-binding pockets (e.g., HCY_8, HCY_29, HCY35, HCY 59, HCY_88, HCY_112, HCY_153, HCY_161), and identification of their amino acid sequences. (ii) Suggestion that some mutations in Delta S1 variant (E156-, F157-, R158G) are likely to affect dexamethasone and cortisol binding to S1. (iii) Suggestion that some mutations in Omicron S1 variant (H69-, T95I, G142-, Y144-, N211-, L212I, S371L, S373P, S375F, S477N, T478K, E484A, T547K) are likely to affect glucocorticoid binding to S1. (iv) Suggestion that some mutations (K417N, E484K, and N501Y) in Beta S1 variant are unlikely to affect dexamethasone and cortisol binding to S1.
3	Cortisol-Acetylcholinesterase conjugate assay, using Cortisol Express ELISA kit (Cayman Chemicals, Ann Arbor, MI, USA); Protein thermal stability assays; GloMelt™ Thermal Shift Protein Stability Kit (Biotium, Fremont, CA, USA); Thermal stability of SARS-CoV-2 S1 assisted by detection with SDS-PAGE; ACE2 Activity Assay Kit (Fluorometric) (Abcam, Cambridge, UK)	(i) Biochemical confirmation that cortisol directly binds to S1. (ii) Dexamethasone and cortisol promote heat-induced unfolding of S1. (iii) Neither cortisol nor dexamethasone compromises ACE2 enzymatic function.
4	SARS-CoV-2 S1 Protein-ACE2 Binding Inhibitor Screening Kit (BioVision, Milpitas, CA, USA)	(i) Combined dexamethasone and cortisol can inhibit the S1-ACE2 interaction more than each individual glucocorticoid alone. For instance, at 1 nM, cortisol and dexamethasone cooperatively reduce S1 binding to ACE2, to levels comparable to 100 nM cortisol or 100 nM S1 polyclonal antibodies. Ten nM dexamethasone + 1 nM cortisol reduces S1/ACE2 binding from 100% to 33% (~77% reduction). At concentrations above 100 nM, the combination of dexamethasone and cortisol does not produce any difference in inhibition compared to dexamethasone or cortisol alone at the same concentrations. (ii) Cocktails of glucocorticoids and a human chimeric monoclonal anti-SARS-CoV-2 S1 antibody cooperatively increase the inhibition of SARS-CoV-2 S1-ACE2 interaction. For example, the mixture of 10 nM cortisol and 100 nM human chimeric antibody inhibits S1/ACE2 binding from 100% to 27% (~73% reduction), well below the levels of 100 nM S1 chimeric antibody alone.

SARS-CoV-2: severe acute respiratory syndrome coronavirus 2; S: spike; GPUs: graphics processing unit; AMBER: accelerated assisted model building with energy refinement; VMD: visual molecular dynamics; MM/GBSA: molecular mechanics generalized Born surface area; RBD: receptor-binding domain; NTD: N-terminal domain; LC-MS: liquid chromatography-mass spectrometry; HCY: cortisol; ACE2: angiotensin converting enzyme-2; SDS-PAGE: sodium dodecyl sulfate-polyacrylamide gel electrophoresis.

## Data Availability

Not applicable.

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
