# Peer review of "Could Endogenous Glucocorticoids Influence SARS-CoV-2 Infectivity?"

_cells, 2022, doi:10.3390/cells11192955_

Round 1

Reviewer 1 Report

The Authors discuss an interesting topic as whether endogenous glucocorticoids may interfere with SARS-CoV-2 infectivity in humans.

Although the manuscript is well built and interesting to a general readership, some ameliorations may improve the article:

1. excess of autocitations troughout the paper may be avoided. It is ok to cite Hassan et al [15], but remarking this reference several times along the text appears excessive.

2. Introduction, line 43: I would say "autoimmunity" instead of citing only "reumatoid arthritis".

3. Page 2, line 57, "purported" is not a commonly used term.

4. Paragraph 2.2 Page 2, lines 91-96: this is a bit obscure and difficult to follows. In general, the Authors may want to ameliorate their tables to better describe each concept, instead of only listing the various studies. The table could accompany the reader throughout the manuscript step by step.

5. Page 3, same as above. Too specific and difficult to follow at some points. I would suggest to summarize most of these infos in the tables (more schematically) and to leave easier statements within the main text.

6. lines 193 to 204, same as above. Try to simplify the concepts for the reader.

7. lines 230 to 240 may be summarized, are not useful and interrupt the flow of reading.

8. paragraph 3.2: long sentences may be avoided. The paragraph is too long and a bit repetitive.

9. lines 265- 268 these sentences report a sort of "finalistic" view of the problem. I would try to expose the hypothesis more clearly.

10. the whole section 3 has been repeated twice. Please delete.

11. Figure 1 should be implemented to show how glucocorticoids do influence the interaction between Spike protein and ACE2 receptor.

12. I would suggest to be more "translational" by adding a separate paragraph on "clinical evidence" of interference of steroid therapy with SARS-CoV-2 infectivity. This may include the description of several papers addressing the incidence and outcome of SARS-CoV-2 infection in patients with systemic autoimmune diseases and autoimmune cytopenias treated with steroids.

5

Author Response

The authors wish to express their sincere appreciation for the very valuable suggestions made by both reviewers. In response, the authors have introduced a large number of edits to the entire manuscript to reduce the number of autocitations, shorten the number of words, eliminate a repeated paragraph, and present a modified figure (which the authors believe, now better illustrates the model). An enclosed version with changes in trackmode has been uploaded.

REV 1:

The Authors discuss an interesting topic as whether endogenous glucocorticoids may interfere with SARS-CoV-2 infectivity in humans.

Although the manuscript is well built and interesting to a general readership, some amelioration may improve the article:

  1. excess of autocitations troughout the paper may be avoided. It is ok to cite Hassan et al [15], but remarking this reference several times along the text appears excessive.

Response: Done

  1. Introduction, line 43: I would say "autoimmunity" instead of citing only "reumatoid arthritis".

Response: Done

  1. Page 2, line 57, "purported" is not a commonly used term.

Response: the word “purported” was replaced by “proposed”

  1. Paragraph 2.2 Page 2, lines 91-96: this is a bit obscure and difficult to follows. In general, the Authors may want to ameliorate their tables to better describe each concept, instead of only listing the various studies. The table could accompany the reader throughout the manuscript step by step.
  2. Page 3, same as above. Too specific and difficult to follow at some points. I would suggest to summarize most of these infos in the tables (more schematically) and to leave easier statements within the main text.

Response to points 4 and 5: Thank you for very valuable comments. The indicated sections as well as the Table have been vastly edited. The number of words in the Table has been reduced while its connection with the text has been improved to facilitate the flow of the manuscript.

  1. lines 193 to 204, same as above. Try to simplify the concepts for the reader.
  2. lines 230 to 240 may be summarized, are not useful and interrupt the flow of reading.
  3. paragraph 3.2: long sentences may be avoided. The paragraph is too long and a bit repetitive.

Response to points 6, 7 and 8: We have thoroughly edited these sections, as indicated earlier.

  1. lines 265- 268 these sentences report a sort of "finalistic" view of the problem. I would try to expose the hypothesis more clearly.

Response: We have touched up our original description of our hypothesis.

  1. the whole section 3 has been repeated twice. Please delete.

Response: Thank you for noticing this major flaw. The issue has been addressed and section 3 has been, additionally edited for clarity.

  1. Figure 1 should be implemented to show how glucocorticoids do influence the interaction between Spike protein and ACE2 receptor.

Response: Thank you for this very valid comment. We have touched up our original Figure in the hope that the model by which glucocorticoids act is now more visual. The edits made are not very vast but hopefully they are sufficient.

  1. I would suggest to be more "translational" by adding a separate paragraph on "clinical evidence" of interference of steroid therapy with SARS-CoV-2 infectivity. This may include the description of several papers addressing the incidence and outcome of SARS-CoV-2 infection in patients with systemic autoimmune diseases and autoimmune cytopenias treated with steroids.

Response: We have seriously considered adding a translational section. However, the problem at hand is too complex. It might be best to leave the suggested addition for a separate paper that addresses the clinical evidence of interference of steroid therapy with SARS-CoV-2 infectivity for a later article. One of many issues noticed by the authors while considering to write the requested section was that the proposed protective actions of glucocorticoids might be important very early following infection. In addition, high viral loads could override any protection afforded by glucocorticoids. High steroids levels could also involve the glucocorticoids receptors and their complex signaling pathways. There are too many variables and we still understand too little about this problem.

Reviewer 2 Report

The manuscript titled “Could endogenous glucocorticoids influence SARS-CoV-2 infectivity?” describes the authors want to discuss the role of endogenous glucocorticoids in the effective response of SARS-CoV-2 infection. The followings are some concerns and comments have been pointed out that the authors may want to consider.

1) The authors used too many lines to focus on describing single literature, for example, lines 71-82 for reference [16], lines 85-106 for reference [17], and lines 107-140 for reference [18]. I’d suggest the authors make necessary modifications to emphasize the authors’ ideas and the across discussion.

2) Lines 205-253 are duplicated in lines 287-335;

3) Table 1: Too many words in the summary table. I’d highly suggest the authors modify Table 1 description to make it more readable.

Author Response

The authors wish to express their sincere appreciation for the very valuable suggestions made by both reviewers. In response, the authors have introduced a large number of edits to the entire manuscript to reduce the number of autocitations, shorten the number of words, eliminate a repeated paragraph, and present a modified figure (which the authors believe, now better illustrates the model). An enclosed version with changes in trackmode has been uploaded.

REV 2:

The manuscript titled “Could endogenous glucocorticoids influence SARS-CoV-2 infectivity?” describes the authors want to discuss the role of endogenous glucocorticoids in the effective response of SARS-CoV-2 infection. The followings are some concerns and comments have been pointed out that the authors may want to consider.

1) The authors used too many lines to focus on describing single literature, for example, lines 71-82 for reference [16], lines 85-106 for reference [17], and lines 107-140 for reference [18]. I’d suggest the authors make necessary modifications to emphasize the authors’ ideas and the across discussion.

Response: Thank you for the excellent comment. We have edited these sections to specifically address the suggestion made by the reviewer.

2) Lines 205-253 are duplicated in lines 287-335; 

Response: Thank you for noticing this. We have corrected this flaw.

3) Table 1: Too many words in the summary table. I’d highly suggest the authors modify Table 1 description to make it more readable.

Response: Thank you for the very valid comment. As earlier mentioned, we have made the Table  more compact.

Round 2

Reviewer 2 Report

The authors made an extensive update. After struggling for several days, I still could not easily follow your revised manuscript. The citations are completely messed up. For example, in line 31, the starting citation is [26, 28]. The references from 1 to 25 are missing. Please provide a clean version for tracking and reviewing the manuscript easily. Thank you.

Author Response

The authors made an extensive update. After struggling for several days, I still could not easily follow your revised manuscript. The citations are completely messed up. For example, in line 31, the starting citation is [26, 28]. The references from 1 to 25 are missing. Please provide a clean version for tracking and reviewing the manuscript easily. Thank you.

RESPONSE: Thank you for the rigorous review. We regret that the order of the references caused confusion. We can confirm that references from 1 to 25 were not missing. Unfortunately, we listed them using an EndNote version that is over 15 years old and does not contain the MDPI citation format. Rather, we used the format for 'Physiological Reviews' which uses brackets [] like MDPI requests, but we failed to observe that the references had been listed alphabetically.

We have corrected this issue by changing the style to that of Circulation, which does not produce brackets but lists the references almost identically to the style requested by MDPI.

We have also used the opportunity to re-read the manuscript. We could not find any major edition issues. However, we made a few minor edits to further clarify the meaning of some lines. Those edits are now highlighted in yellow for clarity (since admittedly reading changes in track mode can be cumbersome).

The authors sincerely appreciate the time spent by this reviewer on our manuscript.